# Advantages and Potential Benefits of Using Organoids in Nanotoxicology

**DOI:** 10.3390/cells12040610

**Published:** 2023-02-13

**Authors:** Varvara G. Nikonorova, Vladimir V. Chrishtop, Vladimir A. Mironov, Artur Y. Prilepskii

**Affiliations:** International Institute “Solution Chemistry of Advanced Materials and Technologies” (SCAMT), ITMO University, 9, Lomonosova Str., Saint Petersburg 191002, Russia

**Keywords:** organoids, nanotoxicology, nanomaterials, in vitro models, 3D cell cultures

## Abstract

Organoids are microtissues that recapitulate the complex structural organization and functions of tissues and organs. Nanoparticles have several specific properties that must be considered when replacing animal models with in vitro studies, such as the formation of a protein corona, accumulation, ability to overcome tissue barriers, and different severities of toxic effects in different cell types. An increase in the number of articles on toxicology research using organoid models is related to an increase in publications on organoids in general but is not related to toxicology-based publications. We demonstrate how the quantitative assessment of toxic changes in the structure of organoids and the state of their cell collections provide more valuable results for toxicological research and provide examples of research methods. The impact of the tested materials on organoids and their differences are also discussed. In conclusion, we highlight the main challenges, the solution of which will allow researchers to approach the replacement of in vivo research with in vitro research: biobanking and standardization of the structural characterization of organoids, and the development of effective screening imaging techniques for 3D organoid cell organization.

## 1. Introduction

The original organoid definition includes the self-assembly of cells into structures characterized by defined topology and functionality [1,2]. In 2014, it was supplemented by Lancaster et al., who defined organoids as “a collection of organ-specific cell types that develop from stem cells or organ progenitors and self-organize through cell sorting…” [3] and spatially restricted lineage commitment, as in vivo. This implies the presence of multiple organ-specific cell types, the ability to recapitulate some specific functions of the organ, and spatial grouping and organization similar to that of an organ. Currently, the main distinguishing features of organoids are their origin from stem cells through self-organization and replication of the key structural and functional characteristics of their in vivo counterparts [4,5,6,7,8]. At the same time, there have been studies where an organoid’s self-organizing structure was modified with the introduction of differentiated cells [9]. However, the main distinguishing features of organoids, which are organo-typical structures and functions, remain unchanged. Existing reviews of in vitro models for nanotoxicology do not cover the aforementioned specificity of organoids [10]. Simultaneously, several successful 3D cell models have recently been developed to test nanoparticles (NPs) [11]. Therefore, we aimed to evaluate the potential of organoids for nanotoxicological studies. At the beginning of the review, we present data on the growth of articles in which organoids were considered as models for nanotoxicology. In the following, we provide examples of successful studies in which organoids have been used as toxicological models. Based on these data, it is possible to identify the main features of organoids that make them promising models for nanotoxicology, such as 3D cell organization, the simultaneous presence of different cell types, differences in cell sensitivity outside or inside an organoid, modeling of tissue barriers and various routes of entry into the body. Finally, we present several successful examples of organoids as nanotoxicology models. Important aspects that require attention include the production of organoids and evaluation of their various parameters. The unification and scaling of these procedures are necessary to obtain a sufficient amount of data for both statistical analysis and machine learning approaches. At the end of this review, we present several suggestions for the successful application of organoids in nanotoxicology.

## 2. Dynamics of Scientometric Indicators of Works, including Studies of Toxicity and Organoids

We analyzed articles related to the use of organoids in toxicological studies. The search was carried out in the PubMed database for all fields using the keywords “organoid”, “toxicity”, and “nanoparticles”. Thirteen articles were published between 2013 and 2020. It is noteworthy that 22 articles were published over the next two years. We excluded articles that were not related to organoids as multicellular structures and evaluated the remaining articles. Owing to their scarcity, the search query was expanded. As shown in Table 1, over the past five years, the growth of articles devoted to toxicological studies in which organoids are used as a test model is commensurate with the growth of organoid-related publications. Simultaneously, research on organoids is increasing its share in toxicity studies. This suggests that replacing classical toxicological models with organoids will expand as organoid creation techniques advance.

## 3. Organoids in Toxicology Studies

Toxicological research generally has broader objectives than nanotoxicological research. For example, preclinical studies require testing for toxicity, drug safety, and inflammatory intestinal disease. This implies the reproduction of not only the histological structure of the organ but also the corresponding changes in macrophage infiltration and altered microcirculation [12].

Table 2 presents some studies performed in the last seven years, which show the potential of organoids for toxicological studies as models of healthy organs.

As shown in Table 2, the assessment of organoid structure and heterogeneity is sometimes not performed [33]. Research is often qualitative [13,16,38,39]. In some cases, only the basic morphological characteristics of the organoid are described (the average diameter or percentage of surviving or differentiated organoids and the proportion of dead cells in the organoid), which may be related to both organoids and spheroids [18,22,24,30].

Several studies have characterized toxic effects in organoids quantitatively, relying on the definition by Lancaster and Knoblich [3] and including a quantitative description of the structures and states of different cell types. For example, different organoid cell types were quantified in [29,36,40,41]. Some studies have considered the structural organization of organoids [32,43]. The concentrations of the tested compounds are usually selected based on their average concentrations in the blood plasma [13,38].

## 4. Routes of Tested Substances’ Administration into Organoids

Several methods can be used to introduce tested compounds into organoids (Table 2). The main route is incubation in the solution (basolateral exposure) (Figure 1). The other route is microinjection directly into the lumen of the organoid using thin-walled glass capillaries (luminal exposure) [46]. These routes of administration have different effects on organoids. For instance, in a study by Hanyu et al. [47], basolateral deoxynivalenol exposure had a more significant impact on intestinal barrier function and stem cells in enteroids than luminal exposure. Immunofluorescence staining of intestinal epithelial proteins, such as E-cadherin, claudin, zonula occludens-1 (ZO-1), and occludin, showed that only basolateral exposure disrupted intestinal epithelial integrity. Basolateral exposure, but not luminal exposure, suppressed the *Lgr5+* stem cell count and proliferative cell ratio [48,49]. Pradhan et al. compared the luminal and interstitial (e.g., basolateral) exposure of enteroids to Shiga toxin [29]. They found that only luminal exposure induced apoptosis. On the other hand, in human intestinal organoids, both administration routes of toxin resulted in apoptosis induction. This phenomenon may be explained by the absence of mesenchymal cells in enteroids. Their necrosis can trigger the release of toxic compounds that cause epithelial cell death. In addition, loss of Wnt production by mesenchymal cells can result in epithelial cell death [29,50]. 

In addition to differences in cellular responses, there is another important aspect of luminal administration. Luminal injections are much more complex than simple incubations in a medium. Additionally, it is difficult to perform identical microinjections when all organoids are different (even when produced according to the same protocol). This has led to inconsistencies in the experimental data. Moreover, even microinjection leads to organoid damage and disruption of internal space integrity. This can lead to leakage of a lumen content and the test substance itself, which turns luminal exposure to basolateral [51]. 

Thus, when choosing from luminal or basolateral exposure, it is necessary to evaluate both the technical difficulties and validity of this particular method of administration. Alternatively, the organoid polarity reversal approach can be used. With this approach, the sequence of histological barriers is changed, and incubation can be used to study both luminal and basolateral exposures [51,52].

Another route of substance administration has been achieved using microfluidic devices for drug delivery under flow conditions that mimic the pulmonary system [48]. In such models, the investigation may focus on three-dimensional cell migration, topography of metabolic activity, and cytotoxicity. Moreover, these models allow researchers to make physiological-related predictions [49].

## 5. Potential Benefits of Using Organoids in Nanotoxicology

### 5.1. 3D Cell Organization

More than 90% of drugs fail in human clinical trials using preclinical models [53]. In some cases, this is associated with incorrect in vivo models [54]. In this regard, patient-derived organoids offer a valuable screening platform for preclinical testing [9]. Due to the spatial organization of multiple organ-specific cell types, organoids have huge potential as a test system for characterizing the permeability of histological barriers [55,56,57,58,59,60], deposition [61], drug metabolism [62,63,64], alteration of intercellular communications [65,66,67,68,69], and intercellular environment reactions [70,71,72,73,74].

Nanotoxicological studies have shown that 3D cell organization can significantly alter their sensitivity to NPs [75]. However, it cannot be argued that the three-dimensional organization always increases or decreases toxic effects [76]. The increase or decrease in toxic effects in 3D cultures depends on the modeled tissue [77]. For instance, Chia et al. reported on the increased resistance of intestinal spheroids to ZnO NPs’ genotoxic effect [78] while Juarez-Moreno et al. found the opposite effect regarding Ag NPs [75].

### 5.2. Cell Diversity of Organoids

Different cell lines vary considerably in their susceptibility to NP actions [79,80,81]. This depends on many factors, including intracellular reactive oxygen species (ROS) level, autophagy activity, or sensitivity to specific elements [82]. However, for adequate signaling, cells must be located close to each other in the same way as inside the body. Organoids make it possible to model such cell connections. The presence of different cell populations organized in 3D structures allows us to speak of compartmentalization in organoids. In comparison with organoids, spheroids do not show any relevant tissue structure; in other words, their structure presents low similarity to the original tissue [83]. Spheroids can be defined as clumps of cells obtained from differentiated cells that aggregate and exhibit some tissue-like structures [84]. Another advantage of organoids is that they support stem cell and progenitor cell cultures and their cell–cell interactions, unlike traditional 2D cultures [85]. 

### 5.3. Ability to Observe Complex Effects

Using organoids allows researchers to visualize and assess many processes that cannot occur in homogeneous 2D cultures or spheroids. Monitoring of the following changes is of great importance in nanotoxicological research because of organ-specific changes in NPs:Protein corona formation [86,87];The ability of NPs to accumulate in certain cell types [21,88,89];The ability of NPs to pass tissue barriers [90,91,92];Different sensitivities to oxidative stress and other toxic effects caused by NPs have been observed in cells of different organs and organ structures [93,94,95].

Based on these features, we highlighted five main advantages and disadvantages of organoids for nanotoxicological research (Table 3).

## 6. Current Progress in Organoid Use in Nanotoxicology

Studies on organoids as test systems for nanotoxicology are scarce, and are presented in Table 4.

In addition to the goals of nanotoxicological studies presented in Table 3, other areas of application of NPs can be considered, for example, the study of NP-organoid structures [114]. In these studies, the organoid cavity was used as a drug delivery vesicle.

## 7. Importance of Nomenclature

As shown in Table 2 and Table 4, organoids with different levels of structural differentiation and a variety of cell types can be used in toxicological research. The similarity of the organoid to the structures of the imitated organ determines the quality of the obtained results. In this regard, it is essential to describe the intact state of the studied organoid structures as well as to precisely follow the terminological nomenclature. For example, a study on colonic organoids was conducted in 2020 [115]. As a technique for reproducing organoids, the authors referred to the work of 2017 [116], which also used the term “colon organoid”. This work, in turn, refers to another work devoted to the technique of reproducing organoids. This is a 2015 work in which, as it turns out, we are talking about “intestinal epithelial spheroids” [117]. It should be noted that in the works of 2020 and 2017, along with the term “colon organoid”, the term “3D organoid culture” is used, which blurs the concept of an organoid. Additionally, in the original article in 2015, the term “organoid culture” was used only once in the introduction concerning the results of 2011 research.

A study by Angireddy et al. [118] used a model of kidney organoids developed by Hendriks et al. [119], demonstrating that this model could reproduce kidney spheroids. The terms “3D organoid model” and “3D organoid culture” were used in a study by Zhang et al. [120] in relation to the model reproducing hepatic spheroids [121].

In our opinion, these inaccuracies are associated not only with the desire to follow scientific trends but also with the cytological approach to organoids. The focus of researchers’ attention is not the structural organ organization of the model but the three-dimensional environment of cells. Currently, the differences between organoids and spheroids are generally accepted. Therefore, inaccuracies in the nomenclature can mislead readers regarding the object on which the testing was performed, thereby reducing the importance of the obtained results.

The solution to the problem demonstrated above could be the creation of “living biobanks”, providing researchers with “true organoids”. Organoids originating from patient-derived induced pluripotent stem cell (iPSCs) have immense potential for development as accurate preclinical models for testing pharmacological or biological interventions. Drugs that function successfully in human organoids have a higher chance of being safe in clinical trials [9].

## 8. Suitability of Organoids for Biomedical Testing

It is suggested that all the organoid characteristics make these models less suitable for high-throughput/high-content screening and can lead to complications in in vitro assays [122]. In contrast, spheroids are more compliant with high-throughput/high-content screening because they are characterized by easy-to-use protocols and scalable methods (co-culture or monoculture) with high reproducibility [122]. In Table 5, we define the locations of the organoids in the biomedical research system.

## 9. Methods for Assessing Organoid Characteristics

In some toxicological studies, organoids are considered one of the methods for confirming the results of 2D cell culture-based studies. In such research, two series of experiments are usually conducted on 2D cultures and organoids. Moreover, a set of cytological techniques is transferred from 2D cultures to organoids [33]. For example, Grabinger et al. compared the chemosensitivity of ex vivo-cultured intestinal organoids and immortalized intestinal epithelial cells (IECs) and found that the IECs were 10–30 times less sensitive to drugs than mini-guts, indicating that the latter can simulate the intestine more closely [123].

Technologies and experimental procedures developed in other model systems can now be applied to human organoid systems, which will accelerate our understanding of human biology and allow us to validate hypotheses and models generated from animal model systems [7]. The potential of an organoid is not only associated with the 3D organization of cells. Therefore, it is necessary to consider the morphological and stereological characteristics of organoid structures that reproduce human organ structures. The number and distribution of NPs within the compartments of the organoid, the ratio, and the activity of different cell populations are also key advantages of organoids over 2D cultures. In our opinion, cytological studies should be supplemented with morphological methods to assess nanotoxicity [124]. This was also evidenced by several toxicological studies conducted on organoids, which successfully demonstrated the complete replacement of animal models with organoids [29,32]. 

Currently, researchers have a wide range of methods to work with organoids, including confocal microscopy, electron microscopy, light microscopy of classical, histochemical, and IHC-stained sections [125], software for morphometric processing of the obtained data [126,127], and methods for the quantitative assessment of the state of specific organoid cells [128,129,130,131]. Owing to these methods, it is possible to assess structural changes in organoids and replace animal models for organ nanotoxicology.

Two types of research, high-content and high-throughput, can be successfully implemented through a comprehensive quantitative assessment of the functional, morphological, cytological, and omic parameters of organoids (Figure 2). An excellent example of high-content research is demonstrating the power of combining single-cell phenotypic analysis with advanced light microscopy to reveal complex morphogenic processes under defined culture conditions [8]. The researchers extensively used single-cell RNA sequencing to comprehensively characterize cell populations and infer phenotypic trajectories of multiple cell types using RNA velocity. This high-dimensional tool predicts the cell state based on the temporal derivative of individual cell gene expression. This unbiased analysis identified specific gene markers of the myriad cell types present in gastruloids. The fluorescent labeling of certain cells allows live imaging to capture specific cell populations as they participate in morphogenesis [8]. An example of high-throughput research can be a combination of methods, including assays based on metabolic activities, cell morphology monitoring, and spheroid size measurement [77].

## 10. Methods of Advanced Organoid Production

Owing to the variety of existing nanomaterials and the need to obtain large amounts of toxicological data, the production of organoids must be simplified. Methods that would allow high-throughput organoid fabrication and screening are required. These issues can be addressed using dynamic flow systems such as microfluidics and organs-on-a-chip. In addition to technical problems, there are also methodological problems described above, such as the inadequate vascularization of organoids [132], an inverted order of cell layers [133], or lack of immune system cells [134]. These problems can also be solved, in whole or in part, by using flow conditions that are close to the real organism.

High-throughput spheroid production can be achieved with the help of bioreactors. However, standard large bioreactors are unsuitable for this purpose. Organoid production requires precise concentration control of the active substances, oxygen, shear stress, etc. Qian et al. circumvented these limitations by creating a mixing system for a standard 12-well plate [135]. In fact, it is 12 separate mini-reactors, each capable of maintaining a specific set of conditions. The authors demonstrated the creation of brain organoids with different regional specificities. The disadvantages include imperfections in 3D printing (used to print blades of stirrers), heterogeneity, and the lack of vascularization of the resulting organoids. 

The problem of organoid heterogeneity is quite essential since this greatly affects the quality of the results. Xue et al. developed a bioreactor to unify the production of retinal organoids [136]. The bioreactor was constructed on a PDMS chip basis with wells connected through a serpent-like channel. The authors performed simulations and showed that, with such pumping of the medium, the formation of bubbles and other inhomogeneities was significantly reduced. This leads to more uniform organoid growth than in the case of mixing each well. Furthermore, this system showed similar results in terms of morphology, maturation, and NADH levels in organoids compared to static conditions. Cai et al. proposed a unique approach that uses acoustofluidics [137]. Compared with common mechanical mixing, they were able to achieve uniform mixing during organoid growth. This is known to be problematic because large organoids can interfere with the stirrer and affect the mixing parameters. This was achieved by constantly monitoring the rotation rate and adjusting it during organoid growth.

However, organoid formation requires several steps to advance their features. Homan et al. showed that flow conditions lead to the vascularization of kidney organoids [138]. This is reflected in podocyte maturity, tubular structure, and gene expression. These organoids are closer to real organisms. The area of the vascular network increased up to five times, and the density of close contacts increased ten-fold. Simultaneously, a clear dependence on the shear rate was observed in the range of 0.04–4.27 mL/min. Berger et al. also reported a positive effect of medium flow on the maturation rate of midbrain organoids [139]. The main effect was associated with a decrease in the size of the necrotic core in the center of organoids compared with static cultivation, even with medium stirring. Computer modeling and in vitro experiments have shown that the oxygen concentration that leads to cell death is less than 0.04 mol/m^3^. Under flow conditions, 1.8% vs. 6.4% of cells were found to be under this O_2_ concentration when cultured in 24-well plates. Jung et al. also reported enhanced maturation of organoids under flow conditions [140]. This was accompanied by an increase in the expression level of albumin and transferrin, as well as the expression of hepatocyte-specific markers MRP2 and HNF4α. The process of organoid maturation was also shorter and was only 5 days. Accelerated organoid formation has also been demonstrated by Sekiya et al. (renal organoids) [141] and Tao et al. (human islet organoids) [142]. In both cases, the approaches were similar. Tao et al. created a porous membrane with a PDMS mask on top. Separate wells were created inside PDMS, and each well supported the growth of a single islet organoid. They found that under flow conditions, the level of E-cadherin expression, which is very important for pancreatic cells, is almost eight times higher than that under static conditions. They also analyzed a significant number of α- and β-cell maturation markers and found that flow conditions positively affect the dynamics of organoid maturation. In addition, organoid functionality (insulin secretion) was higher under flow conditions. Cho et al. also demonstrated a positive effect of the medium flow on the quality of brain organoids [143]. Cerebrospinal fluid (CSF) circulation plays an important role in neuronal structure formation. The authors developed a simple microfluidic setup using a laboratory shaker to simulate a bidirectional flow. They showed that cultivation under such conditions positively affects not only the transport of glucose into organoids and their viability but also leads to less variability in the expression of specific genes. This is important in the context of the reproducibility of studies conducted on organoids.

As shown in Table 2 and Table 4, a typical approach to study toxic effects on organoids is simple incubation in a culture medium containing NPs. This approach does not reflect the real pathways of exposure that occur in the body, particularly with the intravenous administration of NPs. At first, it is associated with insufficient organoid vascularization. In this case, NPs can enter the organoid only through the paracellular or transcellular pathways. This issue can be addressed by enhancing vascularization using the approaches described above. Second, fluid flow can affect the interaction of NPs with organoid surfaces. The standard method of culturing organoids under static conditions using an extracellular matrix reduces the penetration of substances into the organoids [144]. Calculations show that the penetration of oxygen and nutrients into organoids limits their growth and leads to death [145]. For the same reason, it can be expected that the distribution of NPs within an organoid will be limited. Thus, flow systems, such as microfluidics or bioreactors, are required for both the cultivation and testing of NPs on organoids.

## 11. Conclusions and Future Directions

The further development of organoids to better match organs will complicate organoid structure and increase the diversity of cell populations and intercellular interactions. This will require methods for assessing toxic effects, similar to those created for animals but adapted for a bioengineering approach. Using only cytological methods when analyzing such complex structures will not bring researchers closer to replacing animal models. The unification and systematization of existing morphological and cytological research methods, biobanking, and the creation of new methods for reproducing organoids consisting of several cell types will fully reveal the potential of existing models for nanotoxicological studies and improve their quality. Today, many organs and tissues still lack adequate organoid analogs. For example, organoids formed from different part of the female genital tract (FGT) are currently under active development. Numerous reviews were recently published on the topic [146,147,148]. However, no nanotoxicological studies were conducted on FGT organoids obtained from normal tissues. Limited but promising data are available only for cancer FGT organoids [146,149,150]. At the same time, there are diseases for which organoids and nanomaterials are studied separately, for instance, Zika virus infection [151]. Since the Zika virus is the only one from its genus that can infect the body through sexual transmission, FGT organoid models for nanotoxicology could be of particular interest.

Unfortunately, data on the similarities between the toxic effects in organoids and in vivo models are scarce in the literature. At the same time, because organoids are considered a substitute for animal models, it is of considerable interest to draw parallels in nanotoxicological studies in vivo and in vitro using organoids.

The formation of organs in vivo occurs under conditions of constant cell migration [152,153]. In this regard, the further development of organoids as an alternative to in vivo studies requires adding extra elements to the organoids’ compositions.
The combination of organoid culture with bioengineering approaches, such as organoid-on-a-chip, microphysiological systems [11,154,155,156], and multi-tissue organ-on-a-chip [98].Automated 3D bioprinting has the potential for scaling up the production of organoids and tissue constructs. For post-organoid bioprinting, many hurdles need to be solved, including the improvement of bioprinting resolution, shear stress-induced cell damage due to high cell densities, the development of better bioinks for depositing cell aggregates, and effective vascularization techniques. The goal is to combine microengineering and organoid cultivation technologies to mimic a human working model to imitate any disease or for comprehensive drug testing to avoid the burden of human trials [9]. This prompted the development of other approaches. For example, the 3D bioprinting concept uses organoid-forming stem cells as building blocks, which can be deposited directly into extracellular matrices for spontaneous self-organization [157]. Another solution could be the magnetic levitational bioassembly of 3D tissue constructs [158].Gastruloids are 3D aggregates of embryonic stem cells cultured under defined conditions that display axial organization and gene expression patterns mimicking the earliest stages of organism development. Their use allows researchers to recreate structures analogous to those of organs in many ways. This has been demonstrated in the development of the heart and intestinal tube [8].

Another challenge faced by nanotoxicologists is the development of gels and media for the cultivation of organoids that are equally inert to NPs of different compositions, including the formation of protein crowns. For example, gold NPs exhibit significant binding to the HA gel matrix, hindering their transport and uniform distribution within the culture medium [112]. Some NP types, including gold, silver, copper, quantum dots, and some derivatized silicas, react with thiol chemistry. The culture media contain many soluble serum proteins known to adsorb onto gold surfaces, producing a protein corona. Aggregation could also result from divalent cation (e.g., calcium)-induced gold particles and gold-HA gel aggregation [112].

The increase in article citations associated with a complete characterization of toxic changes in organoid structures (Table 2) is consistent with the data in Table 1. Based on these data, we demonstrate that an increase in the number of articles devoted to toxicological studies in which organoids are used as a test model is commensurate with the increase in organoid publications but not related to the total number of toxicity studies. Thus, we assume that the quality of organoid characterization, in its classical definition, has a significant impact on the quality of toxicological studies [3]. 

Summarizing the above data, we formulated questions to be solved by the nanotoxicology of organoids.

Challenge 1. How can structural variability, cellular heterogeneity, and complexity of organoids be assessed?

Challenge 2. How can the organoid production be robotized, automated, or scaled?

Challenge 3. How can organoid nanotoxicology assessments be quantified based on self-reporting genes?

Challenge 4. How can we adapt these challenges to high-content and high-throughput screening to successfully replace routine 2D cell cultures and animal models?

## Figures and Tables

**Figure 1 cells-12-00610-f001:**
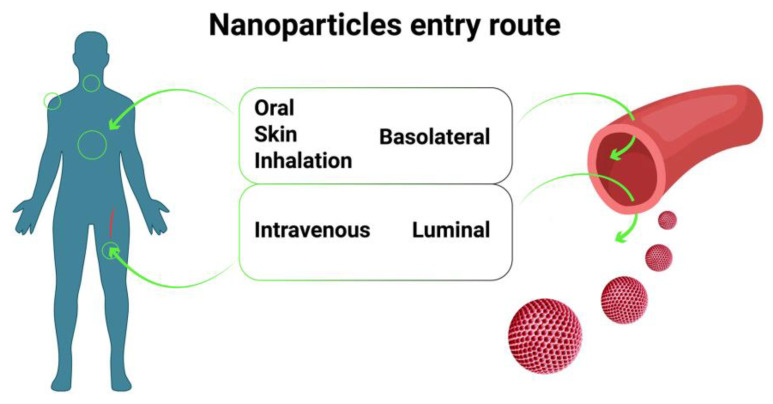
Main routes of NP entry into organisms. Most NPs can enter the body through natural barriers, such as the skin, mucous membranes, or vessel walls. At the same time, to study the biodistribution of NPs, it is important to understand how nanoparticles pass through these barriers. For example, NPs that enter the body through the skin can enter vessels by passing through the basolateral membrane and then through the endothelial layer. If the NPs initially enter the vessel, they can leave it via the luminal pathway, first passing the endothelial layer and then the basolateral membrane in the opposite direction.

**Figure 2 cells-12-00610-f002:**
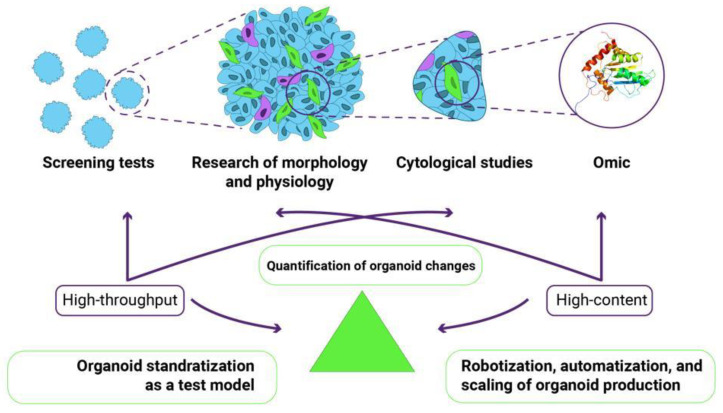
The combination of high-throughput and high-content studies for organoids requires the development and improvement of several methods, namely the automation of organoid production, the standardization of production methods, and the development of standards for assessing morphological changes in organoids.

**Table 1 cells-12-00610-t001:** Distribution of articles by search queries related to toxicology and organoids.

		Year									
		2013	2014	2015	2016	2017	2018	2019	2020	2021	2022 *
Search queries	“organoid” “toxicity” “nanoparticles”	1	1	1	0	1	1	3	5	11	11
	“organoid” “toxicity”	7	16	9	18	37	53	61	132	154	122
	“organoid”	151	189	277	466	761	1033	1360	2141	2839	2113
	“toxicity”	41,075	42,975	44,468	45,950	47,545	49,354	52,021	58,787	62,560	41,896
The share of toxicological studies in the number of articles devoted to organoids		5%	8%	3%	4%	5%	5%	4%	6%	5%	6%
The share of organoids studies in the number of articles devoted to toxicity studies		0.02%	0.04%	0.02%	0.04%	0.08%	0.11%	0.12%	0.22%	0.24%	0.29%

* As of September 2022.

**Table 2 cells-12-00610-t002:** Morphological studies of organoids in toxicological studies in 2015–2022.

Organoid Type	Structures and Cell Diversity of Organoids	Routes of Administration of the Toxic Substances	Visualization and Assessment of Structures and Cell Types	Disadvantages of the Model	Organoid Formation Protocol	Reference
Patient-derived oral mucosa	Small proliferating epithelial cells were located outside, and larger ones with higher differentiation rates inside, the organoid	Incubation for 120 h	Immunohistochemistry (IHC) and hematoxylin-eosin staining	The inverted direction of the histological barrier. The outer layers are proliferating while the inner layers are highly differentiated	[13]	[13]
Human esophagus	Immortalized normal human esophageal keratinocyte cell lines with a differentiation gradient from periphery to center	Incubation for 24 h	High-resolution confocal microscopy (CLSM) and transmission electron microscopy (TEM). An increase in intracellular vacuolar structures has been qualitatively demonstrated	The inverted direction of the histological barrier	[14,15]	[16]
Rat duodenum	Organoids had lobular morphology and formed microvilli lined with intestinal cells, mucus-secreting goblet cells, a small population of enteroendocrine cells, and Paneth cells	Incubation for 24 h	Phase-contrast microscopy was used to quantify the percentage of differentiated organoids over time	Lack of macrophages	[17]	[18]
Mouse/human intestines	Villus-like structures with stem cells, goblet cells, and endocrine cells at the base of the crypt. Cell apoptosis was observed at the tips of the villi	Incubation for 24 h	The percentage of surviving organoids was measured	Lack of macrophages	[19,20,21]	[22]
Mice intestines	Villus-like structures with stem and Paneth cells at the base of the crypt. Cell apoptosis was observed at the tips of the villi	Incubation for several days	Measurement of organoid area and number of buds per organoid	Lack of macrophages	[21]	[23]
Human intestines	Same as above	Incubation for 4 days	Quantification of organoid diameter. IHC staining of different cell types	Lack of macrophages	[21]	[24]
Human liver	Organoids have intraluminal structures; bile canaliculi and pericanalicular sheaths were formed	Incubation for several days	Gene expression analysis, histology examination and IHC staining	Lack of macrophages	[25]	[26]
Human intestines	The epithelial layer contains enterocytes, Paneth cells, enteroendocrine cells, and goblet cells. The epithelial layer is surrounded by mesenchymal cells.	Incubation for several days or injection into the lumen	Visualization and marker expression quantification in epithelial and mesenchymal cells	Deep crypt structures are not seen	[27,28]	[29]
Human alveoli	Organoids had a structure similar to an alveolar sac, with many alveoli and layers of epithelial and mesenchymal cells	Incubation for several days	The organoid diameter was measured	Has the level of fetal maturity	[30]	[30]
Human kidney	Organoids contained kidney tubules subdivided into proximal segments and distal portions, interstitial cells, regions resembling primitive glomeruli with podocytes, and proliferating cells	Incubation once for 24 or 48 h, or four times with a one-day interval	IHC staining	Organoids were composed of immature nephrons	[31]	[32]
Human kidney	The morphology was close to that of normal glomeruli. Extracellular matrix (ECM) was visible within the structure.	Incubation for 48 h	No toxic effects were visualized. The homogenate of organoids was investigated	The tubules of the nephron are missing	[33]	[33]
Human bladder	5–7 cell layers with multiple layers of intermediate cells	Incubation in solution for 60 min	The penetration of particles labeled with a luminescent dye through all layers of the organoid is qualitatively shown	Non-homogenous differentiation with three discrete zones	[34]	[35]
Human testis	Leydig cells, Sertoli cells, spermatogonia, and peritubular cells	Incubation for 48 h	Polymerase chain reaction (PCR) and qualitative CLSM	The organization of cells into structures is not described, although the histological images show the peripheral distribution of Sertoli and Leydig cells	[36]	[36]
Human brain	Immature neurons and astrocytes formed in layers	Incubation for 6 h	TEM and qualitative IHC staining	Has the level of fetal maturity. Neurons migrate to the center of the organoid	[37]	[38]
Human brain	Same as above	Incubation for 24 h	IHC staining for apoptosis and cell proliferation. Assessment of tau and β-amyloid expression.	Same as above	[37]	[39]
Human brain	Dopaminergic neuron spheroid with incorporated astrocytes. Astrocytes were organized radially around the organoid, forming a glial corona	Incubation for 24 h	Quantitative fluorescence staining with calcein and propidium iodide (PI)	There are no data on the spatial distribution of subpopulations of Lund human mesencephalic cells in the organoid	[40]	[40]
Human brain	Large neuroepithelial buds containing fluid-filled cavities. A pool of neural progenitor cells was located near apical surface	Incubation for 10 days	IHC staining	Absence of macrogliocytes and microgliocytes	[41]	[41]
Mouse retina	Continuous epithelial structures with clear stratification, which contain all major neural retina components	Incubation in solution for 2 and 4 days	IHC staining	Absence of hematoneural barrier	[42]	[43]
Human endometrium	Lumen-bordering cell layers. Presence of secretory cells and mucus secretion.	Incubation for 72 h	XTT assay, ion channel activity, Ki67 expression assessment	Absence of endometrial stromal cells, low hormone responses	[44]	[45]

**Table 3 cells-12-00610-t003:** Advantages and difficulties of using organoids in nanotoxicological studies.

#	Organoid Possibilities	Organoid Feature	Challenges	Models that Lack These Features
1	Reproducing intercellular communication and paracrine effects	Organoids are heterogeneous (composed of several types of cells, including stem cells)	Difficulty in standardization and quantification	2D cell cultures and spheroids
2	Investigating the penetration of NPs through tissue barriers	Reproduction of the structural organization of organs	Difficulty in visualizing the penetration process	2D and 3D cell cultures and spheroids
3	Reproducing the reactions of human tissues [7]	Develops from stem cells	Requires the introduction of additional cells into the structure, which during embryogenesis penetrate the tissue by migration	Animal models
4	Reproducing biochemical gradients	Cell nutrition by diffusion	The size of the organoid is limited; the direction of cell differentiation is inverted to the center of the organoid	2D cell cultures
5	The ability to obtain many quantitative data on different cell types and the state of organ structures and functions	Structural complexity	Lack of screening tools for imaging	2D cell cultures and spheroids

**Table 4 cells-12-00610-t004:** Morphological studies of organoids in nanotoxicological studies in 2014–2022.

Organoid Type	Organoid Structures and Cell Diversity	NP Delivery Route	NP Type	NP Dose	Visualization and Assessment of Structures and Cell Types	Organoid Formation Protocol Reference	Ref
Human brain	Multi-layered, neurons expressed cortical layer I, V, and VI markers	Incubation for 24 h	Multi-walled carbon nanotubes (MWCNTs), diameter 5–15 nm, length 0.5–2 μm	16 or 64 μg/mL	Organoids were dissociated into individual cells, stained with a NO probe, dihydroethidium (DHE) superoxide probe, and AO/DAPI to determine cytotoxicity. All results were quantified	[96]	[97]
Human liver	Primary hepatocytes, stellate and Kupffer cells	Incubation for 1 to 2 days	20 nm MgO	100 µg/mL	Quantification of ROS and ATP based on image analysis after IHC staining	[98]	[99]
Human colon	No structural characteristics provided	Incubation for 24 h	10–20 nm SiO_2_ 21 nm TiO_2_	0.8 mM 1.1 mM	Live/dead cell ratio was determined after fluorescent labeling	[100]	[100]
Human pancreatic cancer	No structural characteristics provided	Incubation for 2 or 24 h	Magnetoliposomes with SPION core and phospholipid bilayer. Size 11.1 ± 2.5 nm	225 µg [Fe]/mL	IHC staining, CellTiter Glo Assay for cell viability assessment, apoptotic marker expression measurement	[101]	[102]
Human gut	Cystic structure, consisting of an epithelial cell layer that envelops a hollow lumen. Apical side was covered with mucus	Incubation for 24 h	2 nm gold NPs conjugated with doxorubicin and AlexaFluor 647	50 µg [Au]/mL	Confocal microscopy with AlexaFluor 647, DAPI, and Actin-488	[103]	[103]
Human intestines	Highly convoluted epithelial structures surrounded by mesenchyme	Incubation for 1, 2, and 14 days	50 nm polystyrene	10 and 100 µg/mL	IHC staining, assessment of inflammatory response, TUNEL assay, ROS generation, endocytosis inhibition	[104]	[104]
Human brain	No specific information about inner structure was provided. Neural progenitor cells, neurons, and astrocytes were presented	Incubation for 7 days	PVP-coated 20 nm Ag NPs	0.1 and 0.5 µg/mL	RNA sequencing, IHC staining, TUNEL assay for assessment of apoptosis rates, cytoskeleton structure stability evaluation	STEMdiff Cerebral Organoid Kit	[105]
Mouse intestines	No structural characteristic provided	Animals were subjected to NP action, and organoids were formed from intestines of these animals	10 nm CeO_2_/Mn_3_O_4_ nanocrystals	0.55 mg/kg	The number of organoid crypts was qualitatively determined via light microscopy. Apoptotic cell percent and ROS were qualitatively determined by IHC staining	-	[106]
Mouse intestines	Villus-like structures with stem cells and Paneth cells mixed at the base of the crypt. Cell apoptosis was observed at the tips of the villi	Same as above	~3 nm hydroxylated graphene quantum dots (QDs)	5 mg/kg	The organoid size was determined using light microscopy	[21]	[107]
Mouse intestines	Crypt-like structures fed into luminal domains where apoptotic cells pinched off into the lumen. Epithelial cells formed a monolayer at the organoid-gel interface	Incubation for 3 days	>500 nm Bi_2_Te_3_ nanowires	0, 50, 100, and 200 µg/mL	Quantitative measurements of organoid surface area. Cell viability were quantitatively analyzed based on a modified colorimetric MTT assay	[108]	[109]
Mouse kidney	Structures similar to the proximal tubules of the nephron	Incubations with NMs for 48 h	Gold NPs, size 5.2 ± 1.3 nm G5-OH PAMAM, size 2.6 ± 0.17 nm	56.6 μg/mL and 3.5 μg/mL 0.675 mg/mL and 0.05 mg/mL	A qualitative investigation of IHC-stained sections with biomarkers for kidney toxicity, Kim-1, and TNFα	[110,111]	[112]
Mouse and human kidney	Glomerulus-like structures, podocytes, and proximal tubules had developed in the kidney organoids	Incubations with NMs for 24 h	QDs: CdTe, CdSe/ZnS, InP/ZnS, GO, BP	0, 0.2, 1, 5, and 25 mg/mL	No quantitative assessment has been carried out. Sections stained with hematoxylin-eosin and IHC with antibodies against LTL, NPHS1 or KIM-1 were qualitatively assessed	[113]	[113]

**Table 5 cells-12-00610-t005:** Correspondence of models, toxic effects, and research mechanisms.

Organization Level	Test Object	Toxic Effects	Nanotoxicity Mechanisms	Methods
Cell	2D culture	Cytotoxic effects	Oxidative stress Mutagenic effect	In silico modeling Genetic methods Cytological methods
Tissue	Cell spheroid	Cytotoxic effects	Same as above	Same as above
Organ	Organoid	Influence on the intercellular substance. Permeability of histohematogenous barriers. Cytotoxic effects in disease modeling. Stem cell toxicity	Same as above + interaction with receptors, deposit effects, interaction with the extracellular substance	Same as above + morphological methods
Interorgan integrations	Organ-on-chip	Modifications of nanomaterials and formation of protein corona in the body. Pharmacokinetics parameters	Same as above + toxic effects associated with the formation of the protein corona, effects associated with a combination of toxic effects with diseases	Same as above + physiological methods
Intersystem interactions and reactions of the body	Animal model	Chronic effects and effects on offspring and higher nervous activity	Same as above + Chronic toxicity	Same as above + behavioral tests
Social aspects	Human	Social groups’ lifestyle influence	Same as above + social features of different population groups	Same as above + epidemiology and sociology

## Data Availability

No data was used for this research.

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
