# Peer review of "Advantages and Potential Benefits of Using Organoids in Nanotoxicology"

_cells, 2023, doi:10.3390/cells12040610_

Round 1

Reviewer 1 Report

This up-to-date review paper summarises the advantages and potential benefits of organoids in nanotoxicity testing. I highly evaluate the conference of this manuscript. This review can be a good reference for organoids research. My comments are as follows:

1.       The authors must write the introduction part including the aims and the structure of the review paper.

2.       In my opinion, figure 1 is not clear and so general. The arrows in the right hand are pointed to the same positions. Please make it clearer.

3.       In vitro, in vivo should be italic

Author Response

1. The authors must write the introduction part including the aims and the structure of the review paper.

Thank you for this suggestion, we added conclusive part for the Introduction with review structure.

2. In my opinion, figure 1 is not clear and so general. The arrows in the right hand are pointed to the same positions. Please make it clearer.

We have redrawn Figure 1 for better clarity, and also updated figure caption.

3. In vitroin vivo should be italic

We fixed these issues

Reviewer 2 Report

Thank you for giving me the opportunity to review.

Your review is well organized.

Tables are complicated, can you simplify them more? 

Author Response

Thank you for giving me the opportunity to review.

Your review is well organized.

Tables are complicated, can you simplify them more? 

Thank you for your appreciation of our work! We tried to shorten some text in Tables. 

Reviewer 3 Report

This review highlights the increased interest for using organoids in toxicological studies. 

As known, Organoids  represent a starting model for assessing the cellular/tissue and organ processes associated with the toxic potential of chemical compounds. Since nanotoxicology is  Since nanotoxicology is a very relevant issue for human, animal and environmental health, the   organoids can be useful to set up advanced realistic predictive models.

The review  shows  an accurate analysis of the literature on several topics related to   organoids as useful tool in nanotoxicology.  The authors have referred this point of view to 3D dimension, different cell population. These features are relevant   to observe complex effects of nanoparticles. 

This review  reported also advantages and disadvantages of using organoids  in nanotoxicological studies. The authors have  also reported advantages and disadvantages , which are included as several tables very useful for those who deal with these topics.

Organoids offer the possibility to perform high content and high throughput analysis  which suggest that we are close to being able to replace animal models for organ nanotoxicology, in amore reaistic way than 2D in vitro models.

Author Response

Thank you for your appreciation of our work!

Reviewer 4 Report

Title: Advantages and potential benefits of using organoids in nano-toxicology
This is an innovative and interesting work. Overall, the manuscript is well organized and the
language is appropriate.
This reviewer just provides some suggestions:
- Please, clarify in the text why the authors have chosen to dwell on the use of organoids in
nano-toxicological studies.
- Table 2: I suggest you to include in the cited literature also studies on female reproductive
tract.
- Paragraph 4: Clarify, for the cited routes of substances administration into organoids,
advantages and disadvantages and support the information with more consistent references.
- Figure 1: It is not clear if the figure 1 refers to in vivo routes of nanoparticle entry into the
organism or to the used in vitro routes to insert nanoparticles into the organoids or both.
Clarify in the text and in the figure legend.
- Sub-paragraph 5.2: like for the other aspects described in 5.1 and 5.3 paragraphs, clarify the
influence of the cellular diversity of organoids in nanotoxicological studies.
- Line 131: in the text, refer to table 3.
- Table 4: same of Table 2.
- Line 217-224: join.

Author Response

  • Please, clarify in the text why the authors have chosen to dwell on the use of organoids in nanotoxicological studies.

 Thank you for your question. In our department, we are working a lot with both nanoparticles and 3D cell cultures/histology, and we were always curious about merging both fields. We also noticed that the potential of organoids is not fully exploited in nanotoxicology. This information can be found in the Introduction, Lines 33-38.

  • Table 2: I suggest you to include in the cited literature also studies on female reproductive tract.
  • Table 4: same of Table 2.

Thank you for this suggestion. It is challenging to find a sufficient amount of data about organoids from the female reproductive tract in toxicological research. We have added one work to Table 2. However, we did not find any studies using organoids from the normal tissues of the female reproductive tract in nanotoxicology. Some studies have focused on cancerous organoids; however, in our review, we focused only on normal tissues. Therefore, we have added some suggestions in lines 357–366.

  • Paragraph 4: Clarify, for the cited routes of substances administration into organoids,
    advantages and disadvantages and support the information with more consistent references.
  • - Sub-paragraph 5.2: like for the other aspects described in 5.1 and 5.3 paragraphs, clarify the influence of the cellular diversity of organoids in nanotoxicological studies.

These sections have been updated accordingly.

  • Figure 1: It is not clear if the figure 1 refers to in vivo routes of nanoparticle entry into the organism or to the used in vitro routes to insert nanoparticles into the organoids or both. Clarify in the text and in the figure legend.

We have updated Figure 1 and its caption for better clarity.

  • - Line 131: in the text, refer to table 3.
  • - Line 217-224: join.

We have fixed these and other typos.